# One-Step Synthesis of Nanoliposomal Copper Diethyldithiocarbamate and Its Assessment for Cancer Therapy

**DOI:** 10.3390/pharmaceutics14030640

**Published:** 2022-03-14

**Authors:** Radu A. Paun, Daciana C. Dumut, Amanda Centorame, Thusanth Thuraisingam, Marian Hajduch, Martin Mistrik, Petr Dzubak, Juan B. De Sanctis, Danuta Radzioch, Maryam Tabrizian

**Affiliations:** 1Department of Biomedical Engineering, Faculty of Medicine and Health Sciences, McGill University, 3775 Rue University, Montreal, QC H3A 2B6, Canada; radu.paun@mail.mcgill.ca; 2Research Institute of the McGill University Health Centre, 1001 Decarie Blvd, Montreal, QC H4A 3J1, Canada; daciana.dumut@mail.mcgill.ca (D.C.D.); amanda.centorame@mcgill.ca (A.C.); danuta.radzioch@mcgill.ca (D.R.); 3Division of Experimental Medicine, Faculty of Medicine and Health Sciences, McGill University, 1001 Decarie Blvd, Montreal, QC H4A 3J1, Canada; 4Division of Dermatology, Department of Medicine, Jewish General Hospital, McGill University, 3755 Cote Ste-Catherine, Montreal, QC H3T 1E2, Canada; thusanth.thuraisingam@mail.mcgill.ca; 5Division of Dermatology, Department of Medicine, The Ottawa Hospital, University of Ottawa, Ottawa, ON K1H 8M5, Canada; 6Institute of Molecular and Translational Medicine, Faculty of Medicine and Dentistry, Palacky University Olomouc, Hnevotinska 1333/5, 77900 Olomouc, Czech Republic; marian.hajduch@upol.cz (M.H.); martin.mistrik@upol.cz (M.M.); petr.dzubak@upol.cz (P.D.); juan.desanctis@ucv.ve (J.B.D.S.); 7Czech Advanced Technology and Research Institute, Palacky University Olomouc, Krizkovskeho 511/8, 77900 Olomouc, Czech Republic; 8Faculty of Dentistry and Oral Health Sciences, McGill University, 3640 Rue University, Montreal, QC H3A 0C7, Canada

**Keywords:** liposomes, copper diethyldithiocarbamate, ethanol injection, cancer therapeutics, melanoma, protein corona, drug delivery

## Abstract

The metal complex copper diethyldithiocarbamate (CuET) induces cancer cell death by inhibiting protein degradation and induces proteotoxic stress, making CuET a promising cancer therapeutic. However, no clinical formulation of CuET exists to date as the drug is insoluble in water and exhibits poor bioavailability. To develop a scalable formulation, nanoliposomal (LP) CuET was synthesized using ethanol injection as a facile one-step method that is suitable for large-scale manufacturing. The nanoparticles are monodispersed, colloidally stable, and approximately 100 nm in diameter with an encapsulation efficiency of over 80%. LP-CuET demonstrates excellent stability in plasma, minimal size change, and little drug release after six-month storage at various temperatures. Additionally, melanoma cell lines exhibit significant sensitivity to LP-CuET and cellular uptake occurs predominantly through endocytosis in YUMM 1.7 cancer cells. Intracellular drug delivery is mediated by vesicle acidification with more nanoparticles being internalized by melanoma cells compared with RAW 264.7 macrophages. Additionally, the nanoparticles preferentially accumulate in YUMM 1.7 tumors where they induce cancer cell death in vivo. The development and characterization of a stable and scalable CuET formulation illustrated in this study fulfils the requirements needed for a potent clinical grade formulation.

## 1. Introduction

The disulfiram derivative and active metabolite copper diethyldithiocarbamate (CuET) metal complex was recently reported as a promising candidate drug for cancer therapy. The drug blocks protein degradation by inhibiting the p97-NPL4 complex, which is responsible for shuttling misfolded proteins from the endoplasmic reticulum (ER) to the proteosome. This event leads to ER stress via the accumulation of misfolded proteins, a heat shock response and, ultimately, cell death [1]. CuET could be used as a sensitizing agent in combination with current therapies to improve cancer treatment efficacy as cancer cells tend to accumulate misfolded proteins, resulting in increased protein turnover [2]. For instance, in combination with cytotoxic agents or immune checkpoint inhibitors, CuET could further improve their therapeutic effect, leading to better patient treatment outcomes. However, CuET is practically insoluble in water and exhibits poor bioavailability, which presents a major obstacle in developing a clinically viable therapy. Nanoparticle-based formulations using biocompatible polymers, proteins, inclusion complexes, or lipids have been developed to date, albeit with some potential limitations regarding cost, scalability, efficacy, and toxicity [3,4,5,6,7]. For example, Li et al. developed a protein formulation of CuET using apoferritin as a carrier that could target cancer cells overexpressing the ferritin receptors [8]. However, since ferritin plays an important physiological role, this formulation may lead to off-target delivery [9].

It has been recently shown that nanoparticle transport across tumor blood vessels and their subsequent accumulation at the tumor site, known as the enhanced permeation and retention (EPR) effect, is predominantly resulting from an active cellular transport process mediated by transcytosis across the tumor endothelium [10]. These findings are contrary to the previously proposed ‘leaky vasculature’ model, even though some passive transport may still occur [10,11]. Studies have also shown that nanoparticle targeting across various barriers, such as the blood-brain barrier (BBB), can be effectively facilitated by manipulating the surface protein corona of nanoparticles. These results demonstrate an active cellular transport mechanism and are consistent with previous findings [12,13,14,15,16]. While protein-based nanoparticles can be tailored for improved transport across the tumor vasculature, the same proteins would need to release the drug or be preferentially endocytosed by tumor cells in situ, complicating the design of such carriers. Compared with other nanoscale delivery systems, liposomes are biocompatible and allow the surface adsorption of multiple types of proteins for the a priori establishment of a protein corona that could be tailored to be more tumor specific [17,18]. Additionally, targeting proteins or molecules can also be grafted onto the surface of nanoliposomes, complementing the protein corona for improved specificity and treatment efficacy (i.e., aptamer or antibody-tagged liposomes) [16]. These advantages make nanoliposomes a strong candidate for targeted drug delivery of CuET to tumors. Nonetheless, to our knowledge, a scalable and easy-to-make liposomal formulation of CuET has not yet been described. Wehbe and colleagues proposed nanoscale liposome reaction vessels where CuET was synthesized at the core of the liposomes by first preparing copper-loaded nanoparticles via thin film hydration and then adding diethyldithiocarbamate that would diffuse into the liposomes and react with the copper ions [19]. While the prepared liposomes were stable in solution, the drug quickly dissociated from the nanoparticles when in contact with plasma in vivo. Furthermore, the manufacturing process was tedious and time-consuming since it required multiple freeze-thaw cycles and purification steps [20].

In this study, we aimed at demonstrating that the preformed CuET metal complex can be loaded into a stable nanoliposomal formulation (LP-CuET) using ethanol injection as a one-step method that is scalable and straightforward to implement using existing standards [21,22]. This method can be used to synthesize LP-CuET within the timespan of two hours with fewer purification steps as compared with lengthier and more expensive methods, such as thin film hydration and extrusion, that can take up to two days. We used the Yale University Mouse Melanoma (YUMM) cell lines, a set of C57BL/6 congenic mouse model cell lines replicating the genetic mutations found in human melanomas as our study model [23,24]. Specifically, the cells chosen were the BrafV600E, Pten−/−, and Cdkn2a−/− male melanoma mouse line, YUMM 1.7, and the UV-irradiated YUMMER 1.7 with a higher mutational burden than YUMM 1.7, along with the human melanoma SK-MEL−28 cell line. The mouse macrophage RAW 264.7 cell line was used alongside YUMM 1.7 to study the toxicity and uptake of nanoliposomal CuET. The RAW 264.7 cell line was chosen to model nanoparticle uptake by macrophages in vivo, as macrophages are one of the contributors to nanoparticle clearance from the blood along with scavenger endothelial cells [25]. The tumor-targeting ability and biodistribution of LP-CuET were evaluated in a YUMM 1.7 subcutaneous mouse model using live in vivo fluorescence imaging. Our study showed that LP-CuET is highly stable and it maintains its cytotoxicity against multiple cancer cell lines. In vivo, LP-CuET is well tolerated in mice at 1 mg·kg^−1^, and accumulates in subcutaneous YUMM 1.7 tumors directly, causing cancer cell death. These results serve as an important foundation for more robust safety and efficacy preclinical studies of LP-CuET in the context of cancer therapy.

## 2. Materials and Methods

### 2.1. Materials

Copper (II) Diethyldithiocarbamate was purchased from Tokyo Chemical Industry (Tokyo, Japan). 1,2-distearoyl-sn-glycero-3-phosphocholine and 1,2-distearoyl-sn-glycero-3-phosphoethanolamine-N-[carboxy(polyethylene glycol)-2000] methoxy- or carboxy-terminated (sodium salt) were purchased from Avanti Polar Lipids (Alabaster, AL, USA). Dulbecco’s Modified Eagle Medium (DMEM) and RPMI-1640 were obtained from Thermo Fisher Scientific (Waltham, MA, USA). YUMM 1.7, YUMMER 1.7 and SK-MEL−28 cell lines were generously gifted by Dr. Ian Watson. Fetal bovine serum (FBS) was purchased from Gibco (Waltham, MA, USA). RAW264.7 cell lines were purchased from American Type Culture Collection (ATCC Manassas, VA, USA). Anhydrous ethanol was purchased from Commercial Alcohols (Boucherville, QC, Canada). Phalloidin-AF480 and Hoescht 33342 were obtained from Invitrogen (Waltham, MA, USA), and Cytopainter from Abcam (Cambridge, UK). All other chemicals used in this study were obtained from Millipore Sigma (Burlington, MA, USA).

### 2.2. Preparation of CuET-encapsulated Nanoliposomes

CuET-encapsulated Nanoliposomes (LP-CuET) were prepared using a slightly modified version of the ethanol injection method as shown in Appendix A [21,22]. A lipid mixture containing DSPC/DSPE-PEG_2000_-COOH/Cholesterol/CuET (mole ratio of 2/0.2/1/1) was added to 5 mL of pure ethanol in a closed container and was heated to 50 °C until complete CuET dissolution. The hot ethanol mixture was injected into 45 mL of rapidly stirred ultrapure water at a constant rate yielding a nanoliposomal dispersion at an ethanol concentration of 10% (*v*/*v*). The resulting solution was transferred to a rotary evaporator to remove the ethanol. The nanoparticles were concentrated using tangential flow filtration with a 100 kDa ultrafiltration membrane to reach a lipid concentration of 3 mg·mL^−1^. The solution was then transferred to a centrifuge to pellet the unencapsulated drug or large aggregates at 2000 RCF for 5 min. For cell and animal experiments, the solution was suspended in PBS, filter sterilized (0.2 µm), and stored at 4 °C. Fluorescent liposomes were synthesized in the same manner as above with the addition of 1% (mol/mol) DHPE tagged with sulforhodamine 101. Methoxy (Me) terminated PEG_2000_ was used at the same molar ratio as above. Empty nanoliposomes were used as control.

### 2.3. Physicochemical Characterization

Following the synthesis, the nanoliposomes were diluted 1:5000 and 1:10 for nanoparticle tracking analysis (NTA) and dynamic light scattering (DLS), respectively. NTA measurements were recorded and analyzed using the Nanosight NS300 (Malvern, UK); DLS and Zeta potential measurements were performed using the Brookhaven Zeta-PALS light-scattering analyzer (New York, NY, USA). Fourier-transform infrared spectroscopy was performed using the Perkin Elmer FTIR Spectrum II (Waltham, MA, USA) on lyophilized samples. Absorbance measurements were performed in 96-well plates at various dilutions using the Spectramax i3 (Molecular Devices, San Jose, CA, USA). For transmission electron microscopy, immediately after synthesis, liposomes were drop cast on a carbon-copper grid and stained with uranyl acetate for contrast. The imaging was performed using the FEI Tecnai G2 Spirit Twin TEM (Hillsboro, OR, USA) at a voltage of 120 kV.

### 2.4. Encapsulation Efficiency

The distinct absorbance spectrum of CuET along with its insolubility in water were used to determine the amount of drug encapsulated in nanoliposomes. First, CuET was dissolved in DMSO at known concentrations and a standard curve was obtained by measuring the absorbance at 450 nm using the Spectramax i3 (Molecular Devices, San Jose, CA, USA). After synthesis, the nanoliposomes were diluted at various concentrations in 1× PBS and centrifuged at 2000 RCF for 5 min to remove aggregates or unencapsulated drug. The absorbance was then measured at 450 nm to fit the data with the standard curve of CuET dissolved in DMSO. DMSO and empty nanoliposomes were used as blanks for CuET and LP-CuET, respectively. The encapsulation efficiency (*EE*%) was determined using the following formula:(1)$$\;EE\%=\frac{D_{S}}{D_{t}}\times 100,$$
where Dt is the total amount of drug added during the synthesis and *Ds* is the amount of drug present in the supernatant as determined from the standard curve extrapolation. The loading capacity (*LC*%) was determined using the following formula:(2)$$\;LC\%=\frac{E}{W}\times 100,$$
where *E* is the amount of entrapped drug calculated from the standard curve, and *W* is the total weight of the nanoparticles after lyophilization.

### 2.5. In Vitro Stability

Nanoliposomes were suspended in ultrapure water (UW), 1× PBS, or 50% FBS in Eppendorf tubes (1.5 mL) at a CuET concentration of 100 µg·mL^−1^. The tubes were placed at 37 °C in a shaking incubator for seven days. At specific timepoints, the tubes were centrifuged at 2000 RCF for 5 min and the concentration of encapsulated CuET in the supernatant of each sample was determined using the standard curve with the appropriate blanks as described above. Drug retention in solution (*RE*%) was determined via the following equation:(3)$$RE\%=\frac{D_{f}}{D_{i}}\times 100,$$
where *Di* is the initial amount of encapsulated drug and *Df* is the final amount of encapsulated drug dispersed in solution at a given timepoint. The storage stability of LP-CuET was determined by suspending nanoparticles containing 100 µg·mL^−1^ CuET in ultrapure water, 0.9% saline, 5% sucrose, or 0.9% saline plus 5% sucrose and placing them at 4 °C, −20 °C, and −80 °C for up to 24 weeks. The change in nanoparticle size was determined via DLS and the amount of aggregated CuET was measured via spectrophotometry as described above.

### 2.6. Mouse Plasma and Hemolysis Study

Male and female C57BL/6 mice aged between 12–24 weeks and weighing between 25–30 g were purchased from Charles River and used for this study. Mice were maintained in specific pathogen-free conditions for the entire duration of the study. Mice were used under FACC approved protocol 2017-7946 and all experimental procedures were approved in accordance with the Animal Care Committee of the McGill University Health Center, Montreal, QC, Canada. Blood was harvested from wild-type C57BL/6 mice into Eppendorf tubes (1.5 mL) containing EDTA (50 µL). The blood was centrifuged at 3000 RPM for 10 min and the plasma was collected. The plasma was used to evaluate CuET release after 48 h using equation (2). For hemolytic analysis, RBCs were resuspended in an equal volume of PBS and the washing process was repeated for a total of five times. RBCs (100 µL) was then diluted into PBS (9.9 mL) to yield a 1% (*v*/*v*) solution. The diluted solution (200 µL) was then mixed with various concentrations of LP-CuET nanoparticles and incubated for 1, 2, or 3 h at 37 °C before being centrifuged. Pictures were taken of the tubes and the optical density of the supernatant was obtained at 580 nm using a spectrophotometer. RBCs suspended in PBS and Triton-X114 served as negative and positive controls, respectively. LP-CuET in PBS was used as a blank.

### 2.7. Nanoliposome-Protein Complexes

Liposomes were suspended in solutions of mouse plasma (MP) at a concentration of 50% (*v*/*v*). For sodium dodecyl sulfate-polyacrylamide gel electrophoresis (SDS-PAGE), the protein corona was isolated using ultracentrifugation at 100,000 RCF for 45 min at 4 °C and washed with 1× PBS three times to remove loosely bound proteins. The final pellet was resuspended in RIPA buffer, and the total protein content was measured using the PierceTM 660 nm assay in 96-well plates. A protein solution of 50% MP without liposomes was used as control/blank during the corona isolation process and protein concentration measurement.

### 2.8. Protein Corona Determination via SDS-PAGE

Pelleted nanoliposome-protein complexes were resuspended in 20 µL of RIPA buffer and 20 µL of 2× Laemmli’s buffer was added subsequently. The solution was heated for 5 min at 95 °C. After protein quantification, 20 µL of each sample was loaded on a 4–20% gradient polyacrylamide gel and ran at 150 V for 60 min. The gel was then stained with a 0.1% Coomassie Blue solution and unstained with a 40% methanol/10% acetic acid solution. Pictures of the gel were taken with a Zeiss optical camera (Oberkochen, Germany).

### 2.9. Evaluation of Cellular Toxicity

SK-MEL-28, YUMM 1.7, and YUMMER 1.7 were cultured in DMEM. RAW 264.7 cells were cultured in RPMI-1640. All the media was supplemented with 10% FBS and 1% penicillin/streptomycin; for YUMM 1.7 and YUMMER 1.7 the media also contained 1% non-essential amino acids. The cells were cultured under recommended conditions in an atmosphere of 5% CO_2_ at 37 °C in a fully humidified incubator. Cells were harvested using 0.05% (or 0.25% for RAW 264.7) trypsin-EDTA (Gibco, Waltham, MA, USA). Cell number and viability was determined using 0.4% Trypan blue staining using the Countess II (Thermo Fisher, Waltham, MA, USA). Cellular survival was determined using the Sulforhodamine B assay as previously described with slight modifications [26,27]. Briefly, cells were seeded in 96-well plates at a density of 3000 cells per well and incubated overnight. The cells were then treated at various concentrations with CuET dissolved in DMSO or LP-CuET for 72 h; the concentration of encapsulated CuET was measured from a standard curve, as described above. Cells were fixed with 50% TCA, stained with 0.4% suflorhodamine B, and resuspended in TRIS buffer (10 mM) at a final volume of 200 µL per well. The optical density was measured at 492 nm.

### 2.10. Cellular Uptake Assessment

Relative cellular uptake was measured in YUMM 1.7 and RAW 264.7 cells using fluorescently tagged LP-Control in media with and without FBS. 10^5^ cells were seeded into 12-well plates and incubated in complete media for 24 h. The cells were then washed twice with PBS and fluorescent LP-Control was added to each well with a dilution of 1:10 in media with or without FBS. The cells were incubated for 3 h at 37 °C, after which the supernatant was removed, the cells were washed thrice with PBS, and the fluorescence intensity was measured with a spectrophotometer at an excitation/emission of 589/615 nm. To evaluate the mechanism of cellular uptake of liposomes, 10^4^ YUMM 1.7 and RAW 264.7 cells were seeded in 96-well plates and incubated overnight. The cells were pretreated for up to 30 min with methyl-β-cyclodextrin (20 µM) and/or chloroquine (30 µM); fluorescence liposomes were diluted 1:100 in complete media and added to each well. The cells were incubated for 3 h at 37 °C. Cells were then washed thrice with PBS and the fluorescence was measured using a spectrophotometer as above. To further support the fluorescence measurements of nanoparticle uptake, leftover nanoparticles were measured in spent cell media using the NTA. Cells were seeded in 6-well plates at a density of 0.3 × 10^6^ and incubated overnight in complete or FBS-free media. Then, a fixed concentration of nanoliposomes (~4–2 × 10^9^ NP) was added to the cell media and incubated for 1, 6, and 12 h. The spent media was then removed with a syringe and filtered through a membrane (0.45 μm) to remove cell debris. The media was then diluted and imaged using the NTA. For each experiment, control wells containing cells without nanoliposomes were used to quantify the baseline level of nanoparticles present in the media at each timepoint, which was subtracted from the nanoliposome-treated wells.

### 2.11. Confocal Imaging

A qualitative study of cellular uptake was performed using confocal microscopy. For live cell imaging, YUMM 1.7 and RAW 264.7 cells were cultured in glass-bottom petri dishes and treated with fluorescent liposomes for 6 h and then stained with Hoescht 33342, Cytopainter for nuclear and lysosome visualization, respectively. To compare the uptake between LP-CuET and LP-Control, cells were treated for 3 h with either fluorescent LP-CuET or LP-Control. The cells were then washed, fixed with 4% paraformaldehyde, and stained with Phalloidin-AF480 and Hoeschet 33352 to visualize actin and the nucleus, respectively. Laser intensity and gain were maintained constant during the imaging process for all samples.

### 2.12. In Vivo Biodistribution

YUMM 1.7 cells were harvested using 0.05% trypsin-EDTA solution and after centrifugation were re-suspended in PBS at a final concentration of 1 × 10^7^ cells/mL. Then, 100 μL of cell suspension was subcutaneously injected into the backs of C57Bl/6 mice. After 12 days, tumors reached approximately 500 mm^3^, and tumor-bearing mice were assigned into the YUMM 1.7 group plus vehicle. Non-tumor-bearing mice were assigned to the control group plus vehicle. On day 12, tail vein injection of fluorescent LP-CuET (1 mg·kg^−1^) was performed, while vehicle mice were injected with PBS. Tumor-bearing and control mice were anaesthetized with 5% isoflurane, and imaged at 1 h, 12 h, and 24 h following injection using In-Vivo Xtreme (Bruker, Billerica, MA, USA). Ex vivo organ imaging was performed at 1-h and 6-h post IV injection with fluorescent LP-CuET in non-tumor-bearing mice and in vehicle-treated mice. Ex vivo tumor and organ imaging were performed on YUMM 1.7 tumor-bearing mice at 6 h post-IV injection with fluorescent LP-CuET or vehicle. Mice were euthanized, and the liver, kidneys, spleen, heart, lungs, and tumors were excised and imaged with In-Vivo Xtreme (Bruker). Fluorescence intensity was normalized to vehicle mice. Tumor tissues were fixed in 10% formalin, stained with H&E, and imaged using a light microscope.

### 2.13. Statistical Analysis

All experiments were carried out independently in technical and biological triplicates (*n* ≥ 3). Welsch’s *t*-test, Brown-Forsythe and Welch one-way ANOVA with Dunnett’s test, and two-way ANOVA with Bonferroni’s correction were used to assess the statistical significance between groups at 95% confidence. The data were considered significant when *p* < 0.05 (* < 0.05, ** < 0.05, *** < 0.005, **** < 0.0001). All statistics were performed using the Prism GraphPad 9 software.

## 3. Results

### 3.1. CuET Is Efficiently Encapsulated Inside Nanoliposomes

The modified ethanol injection method described in the experimental section, and outlined in Appendix A, was used for the synthesis of LP-CuET. The process yielded monodisperse nanoliposomes with a mean size of 111.9 ± 2.1 nm (Appendix A), good colloidal stability, and high encapsulation efficiency, as summarized in Table 1. LP-CuET remained adequately dispersed in solution with little aggregation even after three months of storage at room temperature. Nanoliposomes had a CuET to phospholipid ratio of 0.41 (mol/mol), resulting in a loading capacity and encapsulation efficiency of approximately 13% and 81%, respectively, which is akin to previously reported formulations [3,6,19]. Notably, the encapsulation efficiency (EE) was close to 95% prior to centrifugation if CuET was completely dissolved in the organic phase at ≥50 °C prior to injection into aqueous phase. The zeta potential was −56.60 ± 2.32 mV, which is due to the addition of DSPE-PEG_2000_-COOH to the formulation, giving the particles a larger, net negative charge on the surface to improve their colloidal stability.

The nanoliposome size and CuET encapsulation was imaged using transmission electron microscopy (TEM). Negative staining with uranyl acetate allowed the visualization of individual particle morphology, which remained relatively spherical when CuET was encapsulated in direct contrast to the collapsed cup-like structure seen in LP-Control under the vacuum of the microscope (Figure 1). Additionally, unstained CuET-loaded liposomes showed enhanced contrast at the liposomal core, while uranyl acetate poorly stained the center of LP-CuET forming a halo-like structure (insert) and suggesting CuET was encapsulated inside the core of the liposome, consistent with previous literature findings using cryo-TEM [28]. TEM images showed an average diameter of 106.7 ± 44.24 nm and 115.8 ± 52.79 nm for LP-CuET and LP-Control, respectively.

Furthermore, FTIR spectroscopy was performed to ensure no chemical modifications occurred during the synthesis of LP-CuET. Figure 2a shows the FTIR spectra of CuET powder, lyophilized LP-Control, and LP-CuET. The lack of any chemical shift in the spectrum of LP-CuET when compared to LP-Control suggests that the encapsulation occurred with no apparent changes to the chemical bonding of the nanoliposomes. The presence of additional peaks in the LP-CuET spectrum at positions 1505 cm^−1^, 1437 cm^−1^, 1275 cm^−1^, 1221 cm^−1^, 948 cm^−1^, and 400 cm^−1^ coincided with the peaks present in the CuET spectrum alone. The data suggested that CuET encapsulation is a process driven by non-covalent interactions as molecules in solution try to minimize unwanted interactions, leading to CuET being packed within liposomes during the ethanol injection process. Figure 2c shows the absorption spectra of CuET dissolved in DMSO, and LP-CuET and LP-Control in water; 450 nm was chosen as the two curves intersected at that point (i.e., the absorbance of CuET in both solutions was equal, Appendix A). A standard curve was then plotted for LP-CuET after extrapolating the concentration from the standard curve generated from CuET in DMSO. The two curves are compared in Figure 2d. There was no significant difference between the curves (*p* = 0.7715), suggesting that the concentration of CuET in solution can be accurately determined through this method if the appropriate blanks are used.

### 3.2. LP-CuET Nanoparticles Are Stable under Physiologically Relevant Conditions

To study the stability kinetics in biologically relevant environments, the nanoliposomes were suspended in ultrapure water (UW) with minimal salt concentration and 1× PBS or 50% Fetal Bovine Serum (FBS). The zeta potential of nanoparticles changes from –59.26 ± 2.32 mV in UW to −43.30 ± 1.66 mV in PBS and to −32.93 ± 6.45 mV with the addition of FBS (Figure 3a). The polydispersity index (PDI) also increased from 0.101 ± 0.009 in UW to 0.210 ± 0.016 in FBS as various proteins were being adsorbed on the surface of the nanoparticles (Figure 3b). Upon incubation at 37 °C for five days, the size of the nanoliposomes decreased in both PBS and FBS from 99.30 ± 6.14 nm to 66.23 ± 8.27 nm, and from 156.23 ± 2.30 nm to 119.90 ± 13.62 nm, respectively, as measured by dynamic light scattering (Figure 3c). The measurements are consistent with particle aggregation and fusion, as aggregates were pulled out of solution by centrifugation. The aggregation kinetic curves at 37 °C in Figure 3d demonstrate the particle’s multicomponent aggregation kinetics in UW with fast aggregation during the first 48 h and stabilization over a period of one week, resulting in close to 40% of the drug being pulled out of solution during centrifugation. Interestingly, when the nanoliposomes were dispersed in PBS, the kinetics were stabilized at about 20% of particle aggregation, resulting in roughly a twofold improvement in the colloidal retention efficacy when buffering salts were present, even if the zeta potential and the PDI increased. Moreover, LP-CuET was retained in solution when the nanoparticles were incubated in 50% FBS, which could be due to the presence of serum proteins that seem to improve their stability via surface adsorption. This finding was also validated by suspending LP-CuET in mouse plasma, with minimal drug precipitation over 48 h as compared with PBS (Figure 3e). Another important consideration is the toxicity and colloidal stability of negatively charged liposomal surfaces containing CuET with respect to red blood cells (RBC). Figure 3f shows an ex vivo cell lysis experiment with mouse RBCs exhibiting little-to-no hemolysis after prolonged incubation with LP-CuET at 37 °C and no noticeable drug precipitation.

LP-CuET nanoparticle stability assessment was performed at 4 °C, −20 °C, or −80 °C for up to six months in ultrapure water, 0.9% saline, 5% sucrose, or 0.9% saline + 5% sucrose, indicating overall an excellent shelf-life of the formulation (Figure 4). At 4 °C, LP-CuET retained most drug in solution with no significant aggregation after six months of storage when kept in saline, and saline + sucrose. When stored at −20 °C, only LP-CuET stored in saline + sucrose showed no significant changes to their retention or aggregation, while no significant changes were observed for those stored in sucrose at −80 °C, albeit the difference in the retention at six months in sucrose + saline was not large (94.14 ± 2.96% vs. 89.43 ± 3.13%).

Nanoparticle aggregation at −20 °C was noticeable when LP-CuET were stored in most solutions. Particularly, in UW at −20°C and −80 °C, there was significant liposomal aggregation and fusion, resulting in large LP-CuET precipitates. Most notably, at week 24, the liposome aggregates in UW were very large, to the extent that almost all CuET-loaded nanoparticles precipitated out of solution. In contrast, the size of the nanoliposomes did not significantly change in any of the measured groups when stored at 4 °C. The overall stability assessment suggests that LP-CuET can be stored at 4 °C either in saline or saline + sucrose as well as in sucrose + saline or sucrose at −20 °C and −80 °C.

### 3.3. Cellular Uptake and Cytotoxicity in YUMM 1.7 and RAW 264.7 Cell Lines

To investigate the cellular uptake of nanoliposomes, live confocal imaging was performed on both YUMM 1.7 and RAW 264.7 cells after 6 h treatment with fluorescently tagged LP-Control (Figure 5a,b). Distinct puncta were observed in both cell lines, with a stronger fluorescence signal in YUMM 1.7 cells indicating a higher uptake of nanoparticles when compared with RAW 264.7. Additionally, the nanoliposomes co-localized with acidic cellular vesicles in YUMM 1.7 but not in RAW 264.7 cells. This is noted by the red-stained nanoparticles’ overlap with the green Cytopainter, which is used as a dye to specifically stain acidic vesicles like the lysosome. No significant membrane fluorescence was observed with live imaging in either cell line; however, there was more concentrated and noticeable fluorescent spots towards the periphery of RAW 264.7 cells. These results indicate that the cells interact differently with the nanoparticles in solution with respect to their uptake mechanisms and nanoparticle processing. Membrane deposition of the nanoparticles was observed as a continuous contour at the cellular margins (white stars, Figure 5b,c) in both cell lines in fixed samples. Internalization (white arrows) was predominantly observed in YUMM 1.7 cells as distinct intracellular puncta, whereas in macrophages, internalization occurred at peripheral membrane extensions. The nucleus and actin were stained with Hoechst 33342 and Phalloidin-AF488, respectively. To confirm internalization in cellular vesicles, a 2D depth plot was obtained to show the depth distribution of the vesicles inside the cell (Appendix A). The data suggest that the nanoparticles were delivered into cells via different internalization pathways following nanoparticle membrane deposition.

CuET affects cells by inhibiting protein degradation, and since cancer cells typically express mutated or otherwise damaged proteins and produce more reactive oxygen species (ROS), they tend to be more sensitive to proteotoxic stress [29]. A cell survival study was performed to determine the IC50 of cells treated with various concentrations of LP-CuET (individual fitted curves in Appendix A). When treated with LP-CuET, YUMMER 1.7 had a lower IC50 than YUMM 1.7 (55.75 ± 6.20 nM vs. 91.39 ± 4.98 nM, Figure 6a) confirming that CuET retains its cytotoxicity as a nanoparticle formulation. When treated with CuET in DMSO, the same trend was observed (79.22 ± 8.45 nM vs. 103.06 ± 12.45 nM). In addition, when human SK-MEL-28 cells were treated with CuET and LP-CuET, a similar cytotoxicity profile was observed. Additionally, Figure 6b shows that YUMM 1.7 cells were more sensitive to CuET as they reached cytotoxic levels much faster than RAW 264.7 cells when treated with 1 µM LP-CuET for 6 h. These results support the hypothesis that actively endocytosing cancer cells having higher LP-CuET uptake are more sensitive to CuET treatment.

To study the uptake mechanism of the formulation, we pretreated cells with chloroquine (CQ) or methyl-β-cyclodextrin (MßC). CQ is used as a surrogate inhibitor of clathrin-mediated endocytosis through the reduction in gene expression of phosphatidylinositol-binding clathrin assembly protein (PICALM) and the prevention of lysosomal fusion with endocytic vesicles, therefore disturbing normal vesicle trafficking [30,31]. MßC was used as an inhibitor of caveolae and lipid raft-mediated endocytosis, and to some extent clathrin-mediated endocytosis, via its ability to sequester cholesterol [32,33]. Treatment of YUMM 1.7 melanoma cell lines with fluorescent nanoliposomes showed no difference in uptake when pretreated with the individual inhibitors, but there was a significant reduction, albeit not complete abrogation, in the fluorescence intensity when the inhibitors were combined (Figure 6c). On the other hand, RAW 264.7 cells showed no significant difference in the uptake of nanoparticles when treated with the same endocytosis inhibitors, further suggesting that the main mechanism of RAW 264.7 cellular uptake may not be occurring via endocytosis. These results were confirmed using NTA to measure the number of leftover nanoparticles in the supernatant post-incubation (Figure 7d,e) showing more nanoparticles present in the supernatant of YUMM 1.7 cells treated with CQ and MßC than in RAW 246.7 cells.

Knowing that nanoparticles can adsorb a protein corona, effectively giving them a distinct biological identity when administered in vivo, we sought to investigate whether carboxylated PEG nanoliposomes recruit different surface proteins. As shown in Figure 6e,f, PEG_2000_-COOH liposomes have a different surface protein profile when incubated and isolated in 50% Mouse Plasma (MP) compared with PEG_2000_-Me, but similar concentrations. The signal from adsorbed proteins is detectable at different band intensities along the lanes, suggesting a distinct supramolecular assembly of surface plasma proteins, possibly contributing to the distinct uptake kinetics observed in the cell lines.

To further confirm that RAW 264.7 macrophages have a reduced uptake compared with YUMM 1.7, the presence of nanoparticles in the cell media was studied over the timespan of 12 h by quantifying the number of nanoparticles leftover in the media supernatant at 1-, 6-, and 12-h time points using the NTA (Figure 7). The number of nanoparticles present in the media of YUMM 1.7 cells significantly decreased after three hours, while the particles in RAW 264.7 media remained relatively constant. Furthermore, when the cells were treated with nanoparticles in the absence of FBS, uptake in RAW 264.7 cells drastically increased. This was confirmed by treating cells with fluorescent nanoparticles in the absence of FBS showing a large increase in the uptake of liposomes in RAW 264.7 cells, while no difference was observed in YUMM 1.7 as shown in Figure 7f,g. These findings highlight the importance of surface proteins in not only stabilizing nanoparticles in solution, but also directing the particles’ cellular fate. This finding supports the protein corona’s consideration as a design parameter when developing novel nanoparticles.

### 3.4. LP-CuET Nanoliposomes Preferentially Accumulate within YUMM 1.7 Tumors

To assess the biodistribution of LP-CuET, fluorescent liposomal CuET nanoparticles were injected intravenously (IV) in YUMM 1.7 tumor-bearing or control C57BL/6 mice at a CuET concentration of 1 mg·kg^−1^. Vehicle mice were used to normalize the fluorescence signal, having received an equal volume (100 μL) of PBS. Live in vivo fluorescence imaging showed a significantly large accumulation of LP-CuET nanoparticles at the site of the tumor. In control mice, the LP-CuET nanoparticles predominantly accumulated in the abdominal region (Figure 8a), which is likely due to the hepatic accumulation and processing of the nanoparticles. Ex vivo fluorescence imaging of YUMM 1.7 mouse organs at 6 h and 24 h post-injection confirmed that most nanoparticles accumulate in the tumor compared with any other organs, in contrast to control mice, where most nanoparticles accumulate in the liver (Figure 8b,c). There was a complete loss of fluorescence signal in the tumor and any other studied organs within 48 h, suggesting that the nanoparticles are cleared from the mice by that timepoint. The mean fluorescence intensity present in the liver of tumor bearing mice was significantly lower at 6 h post-injection when compared with the tumor tissue (0.305 × 10^3^ ± 0.201 p·s^−1^ vs. 5.904 × 10^3^ ± 1.829 p·s^−1^). However, the mean intensity in the liver of control mice was higher than that of the liver of tumor-bearing mice, but much lower than the tumor intensity at 6 h (2.176 × 10^3^ ± 0.452 p·s−^1^), as shown in Figure 8e,f. No other significant differences in fluorescence intensity were identified post-1 h in any other studied organs, suggesting that the major targeting site of the nanoparticles is the liver or tumor tissues. At 1 h, faint fluorescence was present in the kidneys and spleen, which are other potential sites of nanoparticle accumulation shortly after IV administration. In addition, H&E staining of LP-CuET-treated tumor tissue showed significant granularity when compared with control tissue (Figure 8d), which is consistent with cell death after 24 h, as well as no morphological changes typically associated with acute organ toxicity (Appendix A).

## 4. Discussion

The aim of this study was to develop and characterize a novel formulation of liposomal CuET using a straightforward manufacturing method. Colloidally stable nanoparticles were synthesized by employing ethanol injection using PEG_2000_-COOH as a surface stabilizer. It is worth noting that ethanol injection also enables the synthesis of non-PEGylated CuET formulations, as well as other PEGylated formulations containing terminal amine, methoxy or hydroxy groups with similar size and drug encapsulation characteristics, but different surface charges (data not shown). The obtained nanoparticles have similar properties as compared to other liposomal formulations described in the literature, including FDA-approved liposomal formulations, such as Doxil^®^ and Onivyde^®^ [3,6,19,34]. Initially, PEG_2000_-COOH was used in this formulation in an attempt to increase the loading capacity and stability of the nanoparticles by improving the surface charge repulsion between the particles in solution [35]. While the loading capacity was not higher in COOH-terminated surface groups, as compared with OH or Me groups, there was a noticeable change in the zeta potential in COOH nanoparticles, which could theoretically reduce the rate of aggregate formation (data not shown). At lower temperatures, the surface charge of nanoparticles is not well stabilized in water showing increased nanoparticle aggregation, whereas in a saline- or saline/sucrose-based solution, the ionic interactions seem to help better stabilize the PEG_2000_-COOH surface groups and membranes (Figure 4). However, liposomes stored in ultrapure water and saline demonstrated increased aggregation and fusion at temperatures below 4 °C as the lipid membranes were not properly stabilized compared with those stored in sucrose, a phenomenon that is well documented [36]. This finding suggests that surface charge stabilization of the nanoparticles by the presence of ions in solution leads to decreased nanoparticle aggregation. Overall, there was very good colloidal stability of the formulation at 4 °C, even after six months of storage, bypassing the need of storing the formulation at very low temperatures, which confirms what was previously reported in the literature for liposomal delivery systems [37]. While the long-term stability of nanoparticles was good at temperatures below 4 °C, the lyophilization of the formulation was more challenging, leading to large aggregates and membrane fusion requiring further optimization.

The presence of anionic PEG chains on the surface of nanoliposomes has been shown to recruit a distinct protein corona in plasma that may result in decreased complement activation and clearance by the reticuloendothelial system (RES) [38]. Additionally, decorating nanoliposomes with surface functional groups would facilitate the development of novel nanomaterial designs using crosslinking chemistry to potentially improve the formulation’s targetability [39]. The surface functionalization of the nanoliposomes with a carboxylic acid group recruits a protein corona that is different from the one seen with PEG_2000_-Me groups in mouse plasma (Figure 6). Having a carboxylic acid group present at the particle interface with plasma results in the recruitment of various proteins, which could be explained by the PEG’s interaction with different protein binding motifs [40]. The supramolecular assembly, conformational changes, and orientation of plasma proteins onto the surface of nanoparticles has been shown to influence the particles’ cellular uptake and biodistribution profile in vivo [39]. The assembly of these proteins onto the surface of the nanoliposomes seems to significantly reduce their uptake in RAW 264.7 macrophages, without affecting their uptake in melanoma cells. Interestingly, macrophage scavenger receptors are known to bind various lipoproteins and polyanions, which could explain the higher nanoliposome uptake in RAW 264.7 cells upon the removal of FBS from the media [41]. This would be an important clinical benefit, as a significant issue in translating nanotherapeutics to the clinic involves the rapid clearing of nanoparticles, from the circulation by the RES [42]. A potential personalized approach could involve pre-assembling the surface protein corona with various proteins that can ensure better stability and more optimal targetability [39]. Our findings further support the notion that the surface of nanoparticles can be rationally engineered to improve tissue tropism in targeted drug delivery; however, the engineering design must consider the particle’s dynamic interfacial interactions with the local protein corona (i.e., in plasma or tumor microenvironment), as adsorbed surface proteins can dictate the biological fate of the particles depending on the cellular milieu and transit time [39,43].

The uptake experiments suggest that the nanoparticles are predominantly internalized and delivered through endosomal pathways to the lysosome (larger inclusions) via vesicle acidification (smaller inclusions, Figure 5) in YUMM 1.7 cells. This is consistent with other nanoliposomal formulations where internalization is mediated by endocytosis [44]. Interestingly, internalized liposomes do not overlap with the green Cytopainter in RAW 264.7 cells, suggesting that the nanoparticle cargo is not delivered through vesicle acidification as observed in endosomal maturation and fusion with the lysosome. In macrophages, nanoparticles are likely delivered through the phagosome via phagocytosis, a vesicle system that does not acidify in certain conditions [45,46]. To confirm the uptake results of live cell imaging, and to ensure that encapsulated CuET does not change the uptake profile of nanoliposomes, the morphological differences in both RAW 264.7 and YUMM 1.7 cell lines were compared. We first tested whether the cells can uptake CuET-loaded liposomes in the same manner as empty liposomes since nanoparticle uptake is primarily a surface phenomenon. Confocal imaging of fixed cells provides evidence that both LP-Control and LP-CuET are being internalized in a similar manner (Figure 5), which allowed us to draw parallels between loaded and unloaded liposomes, as only unloaded liposomes were used in the uptake experiments as to avoid introducing bias with CuET’s toxicity. In this experiment, the observed fluorescence intensity in RAW 264.7 cells was less than in YUMM 1.7 using both formulations, suggesting preferential uptake of the liposomes by YUMM 1.7 cells. Both cell types exhibited some fluoresce intensity at the membrane surface after background correction that is likely due to nanoparticle deposition onto, or fusion with, the cell membrane (white stars), which could serve as another delivery mechanism. Surface accumulation of nanoparticles might be higher on YUMM 1.7 cells as they tend to occupy more surface area in contrast to RAW 264.7 cells, likely being an important experimental limitation. While internalized, nanoparticles can be well observed in melanocytes as red puncta (white arrows), whereas macrophages demonstrate membrane bulging at the surface (white arrows), indicating a cellular uptake primarily driven by phagocytosis rather than endocytosis. The lack of complete reduction in fluorescence with the combination of CQ and MβC inhibitors can partially explain the deposition of liposomes onto the surface of the cell membrane leading to liposomal fusion or other uptake mechanisms, such as clathrin or caveolin-independent endocytosis, pinocytosis, or phagocytosis, which melanocytes can also perform [47]. However, it is important to note that (1) CQ and MβC can perturb multiple endocytic pathways, not just clathrin or caveolin-mediated uptake, and (2) nanoliposomes behave differently when studied in vivo since their clearance and biodistribution is highly dependent on a multitude of parameters. In addition, further uptake experiments should be conducted with scavenger endothelial cells, as they are considered an important mediator of nanoparticle clearance from the bloodstream along with macrophages [25].

CuET was shown to be cytotoxic to multiple cancer cell lines in vitro by inhibiting protein degradation via the inhibition of the p97 protein translocation complex. It was recently shown that CuET can cause a conformational lock of the p97-NPL4 complex under oxidative conditions, which are elevated in cancer cells [48,49]. Melanoma cell lines were chosen as a model since cutaneous melanoma is known to have a significant mutagenic burden, primarily due to UV exposure [50]. YUMM/YUMMER 1.7 cells are a good model to evaluate the efficacy of CuET since these melanocytes are genetically engineered to contain specific mutations, as previously mentioned. We also used the human cell line SK-MEL-28 as it is well documented to have an array of genetic instabilities and mutations [51]. The in vitro cellular survival results in melanoma cell lines seemed to indicate that highly mutated cancer cells are more sensitive to CuET-mediated cell death, as both YUMM/YUMMER 1.7 and SK-MEL-28 showed an IC50 in the nanomolar range. This could be of benefit to patients suffering from tumors that have a higher mutagenic burden, proteotoxic stress, or contain elevated levels of ROS, but that are resistant to first-line therapies, such as immune-checkpoint inhibitors.

Biodistribution experiments demonstrated that LP-CuET nanoliposomes preferentially accumulate inside YUMM 1.7 tumors in C57BL/6 mice as early as 1 h post-IV injection and remain focally concentrated in situ for at least 24 h. In contrast, nanoparticles accumulate in the liver in non-tumor-bearing control mice. These results were also corroborated by ex vivo fluorescence imaging of mouse organs at 6 h and 24 h post-injection. Additionally, granularity observed in H&E staining of LP-CuET-treated tumor tissue suggested that CuET is active at the tumor site and can induce cancer cell death. Preferential accumulation of LP-CuET inside tumor tissue due to the EPR effect has been previously shown with various nanoparticle formulations [52,53,54]. It is also possible in this case that active transport of the nanoparticles into the tumor is mediated by specialized endothelial cells that could be responsible for the observed large accumulation [55]. Future studies would need to confirm the presence of these cells in multiple tumor types, including YUMM 1.7, and whether they are involved in nanoparticle transport and retention. Interestingly, minimal-to-no fluorescence signal was observed in the mouse blood at the 6 h timepoint, suggesting that this formulation has a half-life that is lower than previous PEGylated liposomal formulations, supporting the notion that tumor accumulation occurs rapidly post-injection [54]. This effect might in part be attributed to the presence of PEG_2000_-COOH groups onto the surface of the nanoliposomes, as chemical modifications of nanoparticles and drug carriers have been shown to alter their transport and biodistribution profile both in vitro and in vivo [56,57,58]. However, since the nanoparticles are delivered to the target tissues relatively early post-injection, much of the time-dependent clearance by the liver might be avoided in tumor-bearing mice, thus potentially avoiding significant liver toxicity without impacting the formulation’s drug delivery efficacy.

## 5. Conclusions

The study described the straightforward synthesis of nanoliposomal CuET via ethanol injection, a method that is easy to scale and implement using a manufacturing system that contains appropriate quality standards for consistent reproducibility. The generated nanoparticles had a CuET encapsulation efficiency of more than 80% and they were colloidally stable in solution even after six months of storage, making them suitable for clinical use. LP-CuET was biocompatible and stable for hours in plasma, as demonstrated by minimal particle aggregation without any signs of hemolysis. The cytotoxicity of LP-CuET was enhanced using a nanoliposomal formulation in cancer cells, making CuET a potential adjuvant to currently approved therapeutics. The ideal formulation would increase the plasma stability of CuET by decreasing uptake by macrophages and the RES, while specifically targeting the tumor as demonstrated in YUMM 1.7 tumor-bearing mice. Taken together, these results suggests that liposomal CuET can be considered a viable candidate for clinical trials pending further in vivo pre-clinical studies to evaluate toxicity and efficacy in cancer mouse models.

## Figures and Tables

**Figure 1 pharmaceutics-14-00640-f001:**
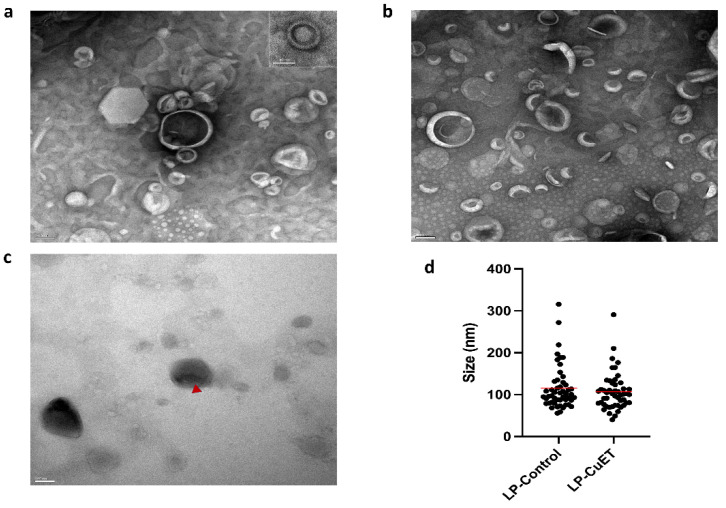
Transmission electron microscopy images of (**a**) LP-CuET and (**b**) LP-Control stained with uranyl acetate. Scale bar: 100 nm; insert: 50 nm. (**c**) Liposomes not stained with uranyl acetate show contrast in CuET-loaded liposomes (◄). Scale bar: 100 nm. (**d**) Size distribution of nanoliposomes. Data are individual points, line is mean, *n* = 50.

**Figure 2 pharmaceutics-14-00640-f002:**
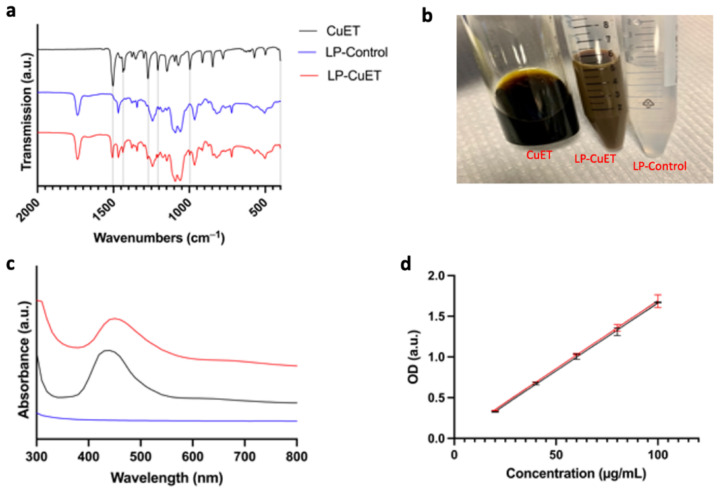
(**a**) FTIR spectra of CuET, LP-Control, and LP-CuET showing no significant chemical shifts, consistent with CuET encapsulation inside the liposome. (**b**) Representative picture of CuET dissolved in DMSO, LP-CuET and LP-Control dissolved in water at equal concentrations right after synthesis. (**c**) Absorbance spectra of LP-CuET, CuET, and LP-Control. (**d**) Standard curves of LP-CuET in water (R^2^ = 0.9950) and CuET in DMSO (R^2^ = 0.9973) plotted as mean ± SD, *n* = 3.

**Figure 3 pharmaceutics-14-00640-f003:**
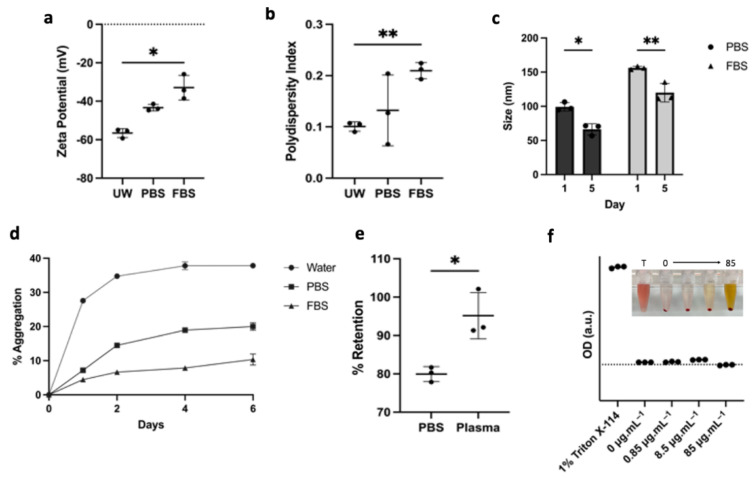
(**a**,**b**) Increase in the zeta potential and polydispersity index of the nanoparticles with the addition of 50% FBS. (**c**) Change in nanoliposomal size after 5 days of incubation at 37 °C. (**d**) Aggregation kinetics of drug-loaded nanoliposomes at 37 °C immediately after synthesis in ultrapure water, PBS, and 50% FBS showing improved stabilization in serum (**e**) Colloidal retention of LP-CuET in PBS or mouse plasma after 48 h incubation at 37 °C. (**f**) RBC hemolysis of various LP-CuET concentrations incubated with RBCs at 37 °C for 3 h. Insert illustrates the retention of LP-CuET in solution at various concentrations post-centrifugation. The data are plotted as mean ± SD, *n* = 3 where * *p* < 0.05, ** *p* < 0.05.

**Figure 4 pharmaceutics-14-00640-f004:**
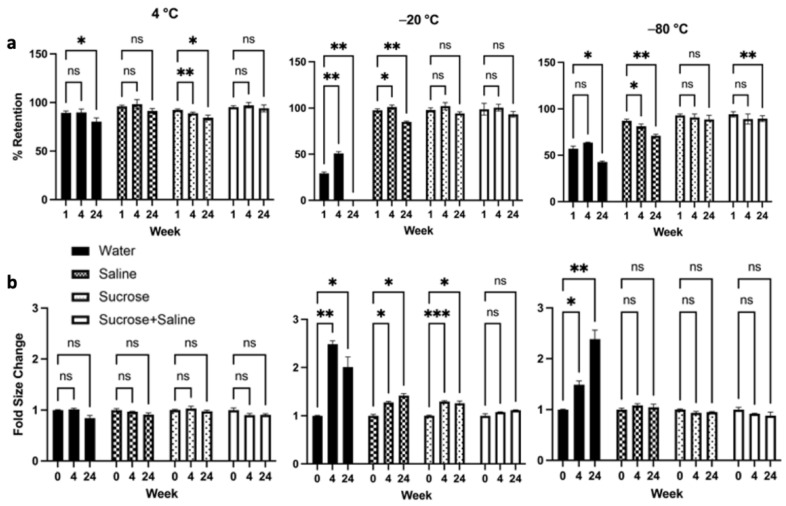
(**a**) LP-CuET retention and (**b**) size kinetics after storage at 4 °C, −20 °C, and −80 °C for six months, showing excellent long-term stability at various temperatures. The data are plotted as mean ± SD, *n* = 3 where ns = not significant, * *p* < 0.05, ** *p* < 0.05, *** *p* < 0.005.

**Figure 5 pharmaceutics-14-00640-f005:**
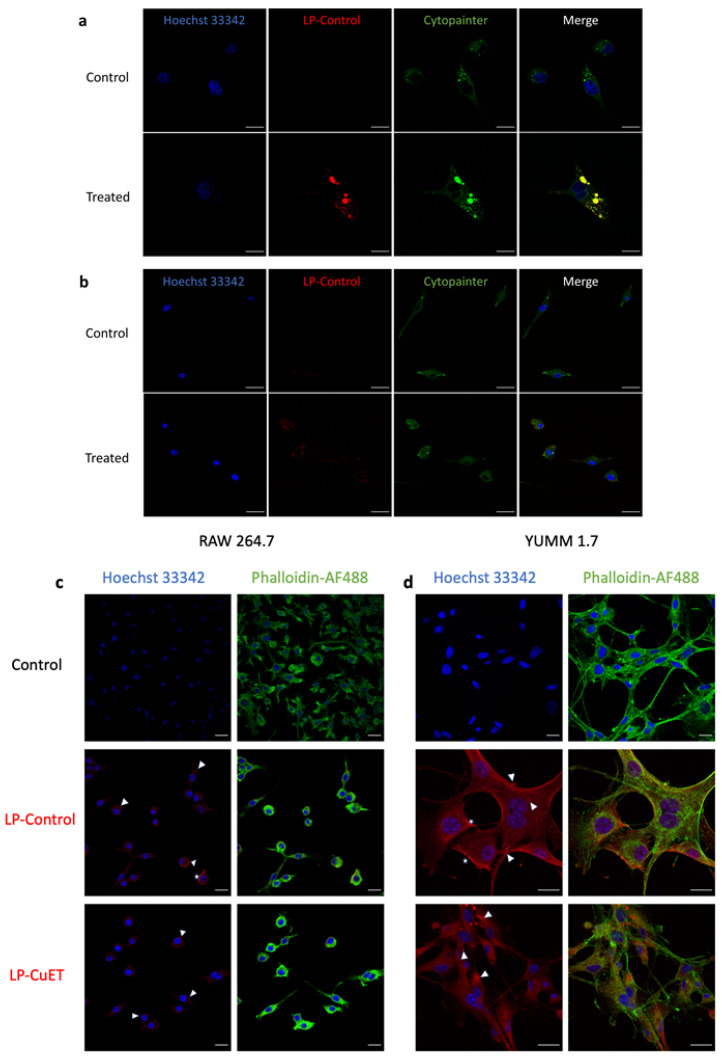
Live confocal imaging of (**a**) YUMM 1.7 and (**b**) RAW 264.7 cells treated with fluorescent LP-Control for 6 h showing multiple colocalization foci of liposomes in acidic vesicles in YUMM 1.7 cells, but not as much in RAW 264.7 cells. (**c**,**d**) Confocal imaging of fixed RAW 264.7 and YUMM 1.7 cells showing similar uptake of both empty and drug-loaded nanoliposomes, including internalization processes, such as phagocytosis or endocytosis (represented by ◁), and cell surface deposition or potential membrane fusion (represented by ✩). Scale bar: 20 μm.

**Figure 6 pharmaceutics-14-00640-f006:**
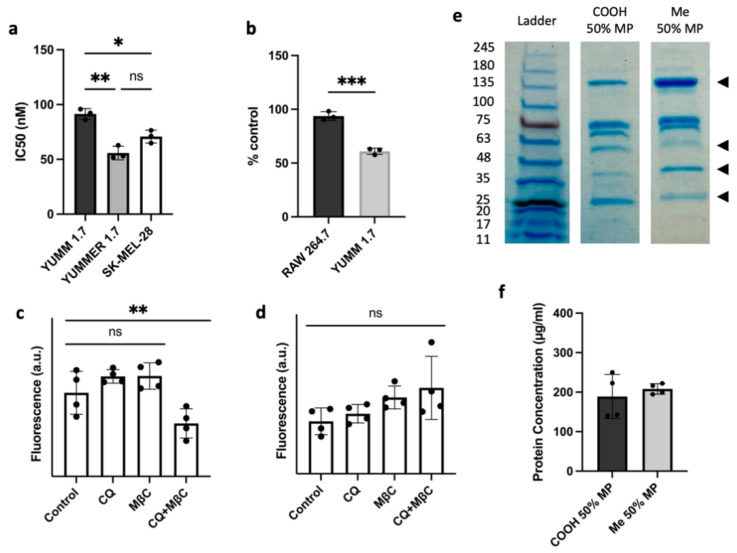
(**a**) IC50 values of melanoma cell lines treated with LP-CuET for 72 h showing that CuET is more effective in killing melanoma cells when delivered as a liposomal formulation. (**b**) RAW 264.7 and YUMM 1.7 cell viability after treatment with 1 µM LP-CuET for 6 h, indicating that YUMM 1.7 cancer cells are more sensitive to the drug. (**c**,**d**) Cellular uptake of fluorescently-tagged liposomes in the presence of the endocytosis inhibitors chloroquine (CQ) and methyl-β-cyclodextrin (MβC) showing a significantly higher uptake of liposomes via endocytosis in YUMM 1.7 cell but not in RAW 264.7, respectively. (**e**) SDS-PAGE gel electrophoresis of carboxy (-COOH) or methoxy (-Me)-terminated PEG_2000_ liposomal-protein complexes where carboxyl-terminated liposomes recruit a distinct protein corona to the surface when incubated in 50% mouse plasma (MP) as denoted by different band intensities (◄). COOH-terminated liposomes seem to recruit less proteins in the 135 kDa range and more proteins in the 63–48 kDa and 25–20 kDa bands. (**f**) Protein concentration post-liposomal-protein corona centrifugation. All data are plotted as mean ± SD, *n* = 3–4 where ns = not significant, * *p* < 0.05, ** *p* < 0.05, *** *p* < 0.005.

**Figure 7 pharmaceutics-14-00640-f007:**
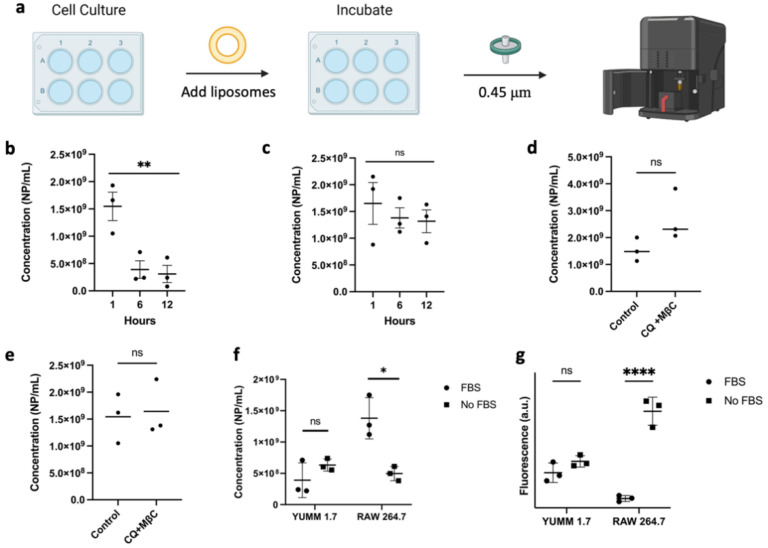
(**a**) Schematic of the method used to quantify nanoparticle concentrations in cell media using the NTA. (**b**,**c**) Concentration of leftover nanoparticles in YUMM 1.7 and RAW 264.7 cultures, respectively. (**d**,**e**) Concentration of leftover nanoparticles in YUMM 1.7 and RAW 264.7 when pre-treated with a combination of chloroquine (CQ) and methyl-β-cyclodextrin MßC. (**f**) The concentration of leftover nanoparticles in cell culture media with respect to the presence of serum proteins after 6 h incubation. Nanoliposomal concentration decreases to a higher extent for RAW 264.7 cells in the absence of serum proteins. (**g**) Cellular uptake of fluorescent nanoliposomes is significantly higher in RAW 264.7 cells when no FBS is present in the media. Data are plotted as individual experiments, mean ± SD, *n* = 3 where ns = not significant, * *p* < 0.05, ** *p* < 0.05, **** *p* < 0.0001.

**Figure 8 pharmaceutics-14-00640-f008:**
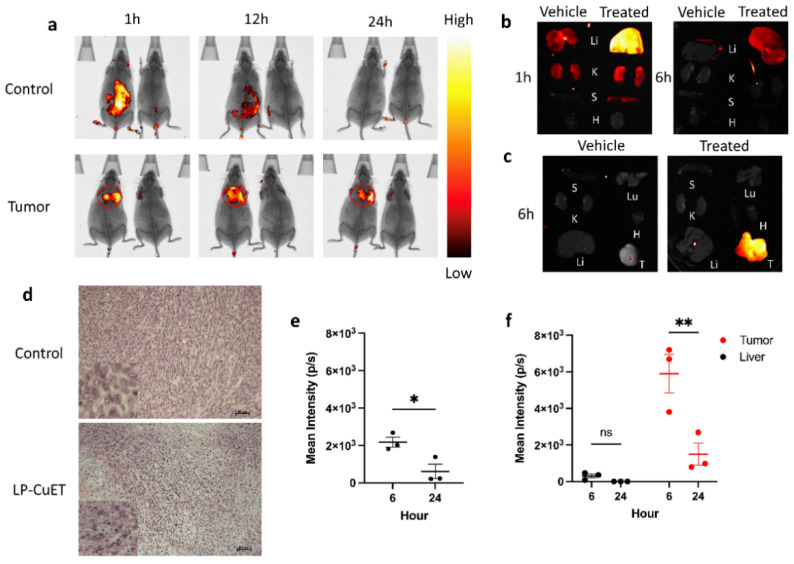
(**a**) Live fluorescence imaging of non-tumor-bearing mice or mice harboring subcutaneous YUMM 1.7 tumors after IV injection with fluorescent LP-CuET. Vehicle mice received an IV injection of PBS and were used to normalize the fluorescence intensity. Treated mice are on the left and vehicle mice are on the right in each image. (**b**) Fluorescence imaging of organs isolated from control mice injected with LP-CuET and compared with vehicle mice at 1 and 6 h. (**c**) Fluorescence imaging of organs and tumor isolated from mice injected with LP-CuET at 6 h. (**d**) Fixed tumor sections isolated from treated mice (1 mg·kg^−1^) at 24 h and stained with H&E and visualized with a light microscope. Inserts are zoomed-in representations. (Scale bar: 50 μm). (**e**) Mean fluorescence intensity in the liver of control mice treated with LP-CuET after 6 h and 24 h post-injection. (**f**) Mean fluorescence intensity of tumors and livers isolated from YUMM 1.7 mice after 6 h and 24 h post-injection. Fluorescence intensity was normalized to vehicle mice in each experiment. Data plotted from individual mice as mean ± SEM, *n* = 3 where ns = not significant, * *p* < 0.05, ** *p* < 0.05. Abbreviations: Li—liver, K—kidneys, S—spleen, H—heart, Lu—lungs, T—tumor.

**Table 1 pharmaceutics-14-00640-t001:** Physical characteristics of LP-Control and LP-CuET nanoliposomes immediately after synthesis. Data represented as mean ± SD, *n* = 3.

Formulation	Size (nm)	Zeta (mV)	PDI	EE (%)
LP-Control	123.5 ± 1.9	−57.36 ± 0.019	0.240 ± 0.019	N/A
LP-CuET	111.9 ± 2.1	−56.60 ± 0.009	0.132 ± 0.069	81.01 ± 1.16 ^1^

^1^*n* = 12.

## Data Availability

Not applicable.

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
