# Peer review of "One-Step Synthesis of Nanoliposomal Copper Diethyldithiocarbamate and Its Assessment for Cancer Therapy"

_pharmaceutics, 2022, doi:10.3390/pharmaceutics14030640_

Round 1

Reviewer 1 Report

The manuscript is focused on developing a delivery formulation for metal complex copper diethyldithiocarbamate (CuET) that is able to induce cancer cell death by inhibiting protein degradation and induces proteotoxic stress, for application in cancer therapeutics.

I have the following concerns/suggestions:

  1. In Figure 1, TEM data do no properly represent the shape of the developed LP,a dn are not of good quality.
  2. How long does it atke for this formulation to be cleared from tumor as well as other organs?

Author Response

Reviewer #1

  1. Comment: In Figure 1, TEM data do not properly represent the shape of the developed LP and are not of good quality.

Response: In the revised version of the manuscript, we have included new TEM micrographs that are more representative of the nanoparticles’ size and shape.

  1. Comment: How long does it take for this formulation to be cleared from the tumor as well as other organs?

Response: Thank you for asking for further clarification. The formulation is cleared within 48h as no more fluorescence signal can be detected at the timepoint. This statement was included in the revised version. It is important to note that fluorescence is an indirect measure of drug clearance, as CuET could still be present at the tumour site past-48h. The amount of drug present at various timepoints will need to be studied more in depth once the detection technique has been optimized and validated. 

Reviewer 2 Report

Overall, the research work in this manuscript is conducted systematically with logical design of the experiments. This article would be attractive to the reader interested in developing a liposomal formulation (of CuET, LP-CuET) for cancer treatment. Although direct therapeutic outcomes of this LP-CuET are not studied here, its targeted accumulation in the tumor seems promising. The data presentation/description needs to improve significantly to convey the findings clearly to the reader. My specifics comments/suggestions for this manuscript are followings-   

1) Remove “..as a Potential Drug Delivery System” from the title. This study is very specific and does not describe any new process that can be used commonly, and there is nothing new in liposomal formulation using ethanol injection.

2) After (or in) Line 85, briefly mention that the preformed CuET complex was loaded into the liposome.

3) Line88-99; instead of mentioning detail about the cells line/mouse model used for this study, authors should mention the primary results, improvement, impact, and therapeutic outcomes from this study. 

4) Line 120, need to mention the mole ratio of the lipids and CuET. How much ethanol was used to dissolve the lipids/CuET mixture? How much ethanoic solution was injected into how much water. What is the total calculated conc. of lipid (gm/mL) in the final liposomal preparation? These are critical info and should be mentioned in this section, specifically when the authors infer large-scale manufacturing.

5) Line 153; Equ (1)-is for EE. Relevant to this, line 289, how loading capacity/efficiency (13%) was calculated? Is the phospholipid concentration determined empirically or calculated based on the fed ratio?

6) Fig4. Sucrose is misspelled in the ligand as “Succrose”

7) Fig5. Caption needs to mention “LP-SR 101” as the control liposome. Neither the caption nor relevant result/discussion (line 369 and so on) tells what Hoechst 33342, cytopainter, and Phalloidin-AF488 staining are for. The authors need to clearly mention those white triangles and stars indicating in the images.

8)Line 408; “…. Endocytosis or pino/phagocytosis”, what does this mean? Do both of these processes occur? How is it confirmed at the point of the manuscript? Overall, the authors need to improve the result/discussion for this section significantly.

 8)Line 503; any plausible explanation of why “…LP-CuET nanoparticles predominantly accumulated in the abdominal region.”

9)Fig8a. Put an outline showing the area for the tumor

10) Fig 8c. It is interesting to see the minimal accumulation of the LP-CuET in the liver of tumor-bearing treaded mice compared with the non-tumor-bearing mice. Why is it so? Is there any other effect than ERP that can be discussed from predominate accumulation of LP-CuET to the tumor?

11) Line 658; “…. good manufacturing practices” what does this mean? “

12) Line 659; “…CuET encapsulation efficiency” how does this calculated? This is related my previous comment.

Finally, any comments on why therapeutic outcomes (such as reducing the tumor volume, survival rate, etc.) are not studied?

Author Response

  1. Comment: Remove “as a Potential Drug Delivery System” from the title. This study is very specific and does not describe any new process that can be used commonly, and there is nothing new in liposomal formulation using ethanol injection.
    Response: Thank you for the suggestion. We have removed “as a Potential Drug Delivery System” from the title.
  2. Comment: After (or in) Line 85, briefly mention that the preformed CuET complex was loaded into the liposome.

Response: The sentence was modified to: “… the preformed CuET metal complex can be loaded into a stable nanoliposomal formulation (LP-CuET) using ethanol injection…”

  1. Comment: Line 88-99; instead of mentioning detail about the cells line/mouse model used for this study, authors should mention the primary results, improvement, impact, and therapeutic outcomes from this study.

Response: Thank you for this recommendation. We have included further detail regarding the primary results, improvements, and impact. Since this study was not focused on the therapeutic outcome of our formulation (efficacy), we did not provide any comment on that matter. We did however specificity that our study serves as an important prelude to preclinical safety and efficacy studies.

  1. Comment: Line 120, need to mention the mole ratio of the lipids and CuET. How much ethanol was used to dissolve the lipids/CuET mixture? How much ethanoic solution was injected into how much water. What is the total calculated conc. of lipid (gm/mL) in the final liposomal preparation? These are critical info and should be mentioned in this section, specifically when the authors infer large-scale manufacturing.

Response: Thank you for bringing this omission to our attention. We have modified the section to contain the proposed modifications: “A lipid mixture containing DSPC/DSPE-PEG2000-COOH/Cholesterol/CuET (mole ratio of  2/0.2/1/1) was added to 5 ml of pure ethanol in a closed container and was heated to 50°C until complete CuET dissolution.” We also included more detail about the post-processing of the formulation for cellular and in vivo studies: “For cell and animal experiments, the solution was suspended in PBS, filter sterilized (0.2 µm) and stored at 4°C.”

  1. Comment: Line 153; Equ (1)-is for EE. Relevant to this, line 289, how loading capacity/efficiency (13%) was calculated? Is the phospholipid concentration determined empirically or calculated based on the fed ratio?

Response: The loading capacity was calculated empirically, and we added the following clarification: “The loading capacity (LC%) was determined using the following formula: (LC% = E/W*100), where E is the amount of entrapped drug calculated from the standard curve, and W is the total weight of the nanoparticles after lyophilization.”

  1. Comment: Sucrose is misspelled in the ligand as “Succrose”.

Response: Corrected.

  1. Comment: Caption needs to mention “LP-SR 101” as the control liposome. Neither the caption nor relevant result/discussion (line 369 and so on) tells what Hoechst 33342, cytopainter, and Phalloidin-AF488 staining are for. The authors need to clearly mention those white triangles and stars indicating in the images.
    Response: Thank you for this recommendation. The figure and caption were updated for clarification. While the staining details are mentioned in the methods section of the paper, we further included more details in the Results section of the paper.
  2. Comment: Line 408; “…. Endocytosis or pino/phagocytosis”, what does this mean? Do both of these processes occur? How is it confirmed at the point of the manuscript? Overall, the authors need to improve the result/discussion for this section significantly.

Response: Thank you for this recommendation. Endocytosis is likely the dominant uptake mechanism in YUMM 1.7 cells as liposome uptake was significantly decreased when treated with endocytosis blockers. However, phagocytosis seems to be the major uptake mechanism in RAW 264.7 macrophages from the analysis of confocal images. The revised version of the manuscript contains clearer information/discussion on this matter.

  1. Comment: Line 503; any plausible explanation of why “…LP-CuET nanoparticles predominantly accumulated in the abdominal region.”

Response: The abdominal fluorescence is probably due to the signal originating from the liver accumulation and processing of nanoparticles. The processing might involve the secretion of some particles in the intestines [Zhang, YN.; et al. Journal of Controlled Release 2016, 28, 332-348].

  1. Comment: Put an outline showing the area for the tumor.

Response: The outline was added in the revised version.

  1. Comment: Fig 8c. It is interesting to see the minimal accumulation of the LP-CuET in the liver of tumor-bearing treaded mice compared with the non-tumor-bearing mice. Why is it so? Is there any other effect than ERP that can be discussed from predominate accumulation of LP-CuET to the tumor?

Response: This a very interesting point. The most straightforward explanation is the EPR effect, however, recent reports argue in favour of the presence of specialized endothelial cells at the tumor site that direct the active transport of nanoparticles into the tumor [Kingston, B.R.; et al. ACS Nano 2021, 15, 14080-14094]. This point is discussed in the new version of the manuscript. It would be an interesting area of investigation for future studies of nanoparticle delivery to tumors.

  1. Comment: Line 658; “…. good manufacturing practices” what does this mean?

Response: The sentence was modified as follows “… a manufacturing system that contains appropriate quality standards for consistent re-producibility.”

  1. Comment: Line 659; “…CuET encapsulation efficiency” how does this calculated? This is related my previous comment.

Response: This is empirically calculated using equation #2 in the methods section.

  1. Comment: Finally, any comments on why therapeutic outcomes (such as reducing the tumor volume, survival rate, etc.) are not studied?

Response: Very accurate and relevant question. This aim of this study was to demonstrates the feasibility of using ethanol injection as an appropriate method for the synthesis of an insoluble anticancer LP-CuET drug delivery system with high encapsulation efficiency, and to investigate its safety and biodistribution in vivo. For a more in-depth evaluation, the therapeutic efficacy of this formulationm is a part of a multidisciplinary research project undertaken by authors using multiple models of cancer.

Reviewer 3 Report

In their manuscript "One-Step Synthesis of Nanoliposomal Copper Diethyldithiocarbamate and its Assessment as a Potential Drug Delivery System for Cancer Therapy"  the authors describe the development and characterization of a stable and scalable CuET formulation needed for a potent clinical grade formulation.

The manuscript is well written - however, with respect on future clincal application (as mentionend by the authors), I would like to aska fpor some more experimental detail:

  • how was proved, that (e.g.) DMSO is really removed ? which analysis was preformed to proof this?
  • - about the sterility of the admistered formulations: How is it guaranteed, thta the dosis injected are sterile ("manufactoring process) ?
  • A short comparison of other nanocarriers with similiar mechanism, based on other matrices (e.g. : such as "The Effect of Chemical Modifications of Chitosan on the Intestina Permeability and Oral Bioavailablity of Carbamoylphosphonate" Journal of Bioequivalence & Bioavailability, no. 2 (2020): 1-8 ; and  some others.... )  in this field; would enhance the quality of the manuscript and provide the reader a better understanding of the preformed research and provide some more information about the background.

Author Response

  1. Comment: How was proved, that (e.g.) DMSO is really removed? which analysis was preformed to prove this?

Response: We apologize for the confusion. Please note that the DMSO was just used for the preparation of standard curve (defining the CuET concentration) as presented in Figure 2.d. No DMSO was used during the encapsulation of CuET in liposome. As described in Materials and Methods section, for the preparation of the formulation, desired concentration of CuET was added to the lipid phase (lipid + ethanol and CuET) prior to the formation of the nanoliposome. The ethanol was then removed using rotary evaporation under high vacuum, leaving behind the CuET-loaded nanoliposomes in the rotary evaporation flask.

  1. Comment: About the sterility of the administered formulations: How is it guaranteed, that the doses injected are sterile (manufacturing process)?

Response: The formulation was filter-sterilized (0.2-micron membrane, see answer to Reviewer #2, comment 4). For GMP scale-up, the sterility of the product can be guaranteed in an appropriate ‘clean’ facility using quality controls; additionally, other methods, such as UV sterilization could also be performed.

  1. Comments: A short comparison of other nanocarriers with similar mechanism, based on other matrices in this field would enhance the quality of the manuscript and provide the reader a better understanding of the preformed research and provide some more information about the background.

Response: Thank you for your suggestion. A short paragraph along with three references on the effect of chemical modifications of drug carriers was added in the discussion section of the revised manuscript.

Round 2

Reviewer 1 Report

I thank authors for addressing my concerns